# Prognostic Relevance of *NPM1* and *FLT3* Mutations in Acute Myeloid Leukaemia, Longterm Follow-Up—A Single Center Experience

**DOI:** 10.3390/cancers14194716

**Published:** 2022-09-28

**Authors:** Erika Borlenghi, Chiara Cattaneo, Diego Bertoli, Elisa Cerqui, Silvana Archetti, Angela Passi, Margherita Oberti, Tatiana Zollner, Carlotta Giupponi, Chiara Pagani, Nicola Bianchetti, Chiara Bottelli, Samuele Bagnasco, Margherita Sciumè, Alessandra Tucci, Giuseppe Rossi

**Affiliations:** 1Department of Haematology, ASST Spedali Civili, 25100 Brescia, Italy; 2Diagnostic Department, Clinical Chemistry Laboratory, ASST Spedali Civili, 25100 Brescia, Italy

**Keywords:** acute myeloid leukaemia, high dose cytarabine (HD-ARAC), idarubicin, *FLT3* mutation, *NPM1* mutation

## Abstract

**Simple Summary:**

In acute myeloid leukemia, molecular genetics abnormalities, particularly *NPM1* and *FLT3* mutations, have a recognized prognostic role in the ELN risk classification and guide treatment decisions, especially since the availability of *FLT3*-targeted drugs. The NILG-AML 01/00 protocol uses a modified induction approach (ICE) and delivers the most active cytostatic agents at maximal doses in consolidation (a high dose of ARAC plus idarubicin) with autologous stem cell support. It calls for allogeneic transplant only in ELN high-risk patients including *NPM1*-wt *FLT3*-ITD-mutated patients. The results obtained from 171 patients showed that the median survival was not reached, and 5y-OS was 58% +/− 4. The prognostic influence of co-mutated *FLT3* was overcome, and the efficacy of this treatment reduced the need for early consolidation with an allogeneic transplant in double-mutated patients, in their first complete remission. These data could represent the benchmark against which results of therapeutic programs using second-generation FLT3-targeted drugs should be compared.

**Abstract:**

The prognosis of acute myeloid leukemia depends on genetic aberrations, particularly *NPM1* and *FLT3*-ITD mutations. The targeted drugs’ availability has renewed interest in *FLT3* mutations, but the impact of these genetic alterations using these treatments is yet to be confirmed. Our objective was to evaluate the results obtained with the intensified NILG-AML 01/00 protocol (ClinicalTrials.gov Identifier: NCT 00400673) in 171 unselected patients (median age, 54.5 years, range 15–74) carrying the *FLT3* (ITD or TKD) and/or *NPM1* mutations. The CR rate and 5-y survival were 88.3% and 58% +/− 4, respectively, significantly higher in the *NPM1*-mutated (CR 93.9%, *p*: 0.0001; survival 71% +/− 6, *p*: 0.0017, respectively). In isolated ITD patients, the CR was lower (66.7%, *p*: 0.0009), and the 3 years-relapse-free survival worse (24%, *p*: <0.0002). The presence of ITD, irrespective of the allelic ratio, or TKD mutation, did not significantly affect the survival or relapse-free survival among the *NPM1*-co-mutated patients. Our data indicate that a high dose of ARAC plus idarubicin consolidation exerts a strong anti-leukemic effect in *NPM1*-mutated patients both with the *FLT3* wild-type and mutated AML, while in the *NPM1* wild-type and *FLT3*-mutated, the therapeutic effect remains unsatisfactory. New strategies incorporating target therapy with second-generation inhibitors will improve these results and their addition to this aggressive chemotherapeutic program merits testing.

## 1. Introduction

Acute Myeloid Leukemia (AML) is a heterogeneous disorder characterized by a wide range of cytogenetic and molecular aberrations [1], which influence prognosis and can guide the choice of treatment [2].

Mutations of *Nucleophosmin-1* (*NPM1*) and activating mutations in *FMS-like tyrosine kinase 3* (*FLT3*), including internal tandem duplications (ITDs) and tyrosine kinase domain (TKD) mutations represent the most frequent genetic aberrations in AML [1,3,4,5]. While *NPM1* mutation is associated with a good prognosis [6,7,8], *FLT3*-ITD confers a poor prognosis, even after intensive chemotherapy and/or stem cell transplant [9,10,11,12].

The concomitant presence of *NPM1* mutation reduces the negative prognostic impact of *FLT3*-ITD, which is also modulated by the *FLT3*-ITD/wild-type (wt) allelic ratio [13,14,15]. Mutations within TKD are the second most common type of *FLT3* mutation in AML, occurring in up to 14% of adult patients with AML [16,17]. They did not, apparently, play a negative role in the prognosis of the overall AML population; however, their impact is still debated and may depend on additional mutations, as well as on the cytogenetic background [16].

With the advent of *FLT3* inhibitors, the management and prognosis of this subgroup of AML patients have changed. Multiple small molecule TKIs that target *FLT3* have demonstrated clinical activity as single agents or in combination with chemotherapy, as reported in the randomized AML Trial RATIFY trial, which documented the efficacy of midostaurin in combination with intensive treatment in adult AML patients aged <60 years [18,19,20].

In this study, we analysed the very long-term impact of the presence of *NPM1* and/or *FLT3* mutations in a large cohort of AML patients, homogeneously treated in the pre-midostaurin era with a program different from classical 3+7, whose overall intensity was higher. It consisted of the use of idarubicin and etoposide in association with cytarabine (ICE) as the induction and of idarubicin in association with High-Dose (HD) ARAC as consolidation therapy and used allogeneic transplant (allo-SCT) only in *FLT3*-ITD-mutated (m) patients without the *NPM1* mutation. Considering the outstanding long-term results obtained without *FLT3* inhibitors, evaluation of the addition of *FLT3* inhibitors also to chemotherapy programs different from classical 3+7 is needed to further improve the outcome of *FLT3*-mutated AML.

## 2. Methods

### 2.1. Study Design and Endpoints

This retrospective study analysed a total of 366 “de novo” AML patients, diagnosed between 2005 and 2021 and considered fit for intensive treatment managed at our institution with the NILG AML-01/00 risk-oriented protocol (ClinicalTrials.gov Identifier: NCT 00400673). Patients with acute promyelocytic leukemia were excluded. 

The present analysis was focused particularly on evaluating the prognostic role of *FLT3* and *NPM1* mutations within an unselected AML population, homogeneously treated, in terms of response to treatment, relapse-free survival (RFS), and overall survival (OS).

### 2.2. Diagnostic Workup and Molecular and Cytogenetic Analysis

The diagnosis of AML was made on stained bone marrow aspiration/biopsies according to ELN criteria [2].

Cytogenetic results were described according to the International System for Human Cytogenetic Nomenclature [21]. Karyotype risk classification was performed according to MRC [22]. The presence of *NPM1* and *FLT3* mutations was evaluated by PCR as described by Gorello, et al. [23] and by Nakao et al. [24], respectively. The *FLT3*-ITD allelic ratio and minimal residual disease (MRD) evaluation in *NPM1*-mutated (m) patients were performed on stored samples in patients diagnosed before 2017, according to ELN recommendations [25]. As the period of observation is very long (17 years), we did not perform a comprehensive genetic study on all the patients at the beginning of the study.

### 2.3. Treatment Plan, Risk Assessment, and Criteria of Response

The treatment protocol (NILG AML-01/00 study; NCT00400673) is shown in Appendix A and detailed in Appendix B.

The main differences compared to standard daunomycin+cytarabine 3+7 induction and HD-ARAC consolidation were the addition of etoposide in the first induction cycle, the use of idarubicin instead of daunomycin in induction, the mobilization of CD34+ peripheral blood stem cells (PBSCs) with a short HD-ARAC course, and the delivery of three consolidation cycles with maximal doses of ARAC, which included also the addition of idarubicin. To minimize haematological toxicity due to HD-ARAC (A20 courses), the patients received the reinfusion of a limited amount of CD34+ PBSC (1–2 × 10^6^/Kg) after each course.

Allo-SCT was planned in patients with an *FLT3*-ITD-isolated mutation and adverse karyotype or late responder (CR after 2 courses) in patients with a persistent molecular disease detectable at the end of the entire treatment program and in patients with haematological or molecular relapse. 

Response evaluation was performed between days 28 and 35 following the start of induction. Complete remission (CR) was defined as a reconstitution of normal marrow cellularity with <5% blast cells and more than 1 × 10^9^/L neutrophils and at least 1 × 10^11^/L platelets in the peripheral blood. Refractory AML was defined as less than CR and persistence of >5% AML blasts in the bone marrow and/or in peripheral blood. Relapse was defined as the reappearance of leukemic blasts in the peripheral blood, recurrence of >5% blasts in the bone marrow, or appearance of extramedullary leukaemia. 

### 2.4. Statistical Analysis

The overall survival (OS) was calculated from the date of diagnosis to the date of death of any cause. The relapse-free survival (RFS) was calculated from the date of CR until the date of the first haematological or molecular AML relapse or the patient’s death.

Survival was evaluated using the Kaplan–Maier method and compared using log-rank tests. The statistical significance threshold was set at a *p* value < 0.05.

Fisher’s exact test was applied to compare dichotomous variables.

Variables found to be significant in univariate analysis were tested in multivariate analysis by the Cox proportional hazard regression model. Analysis was performed with SPSS version 22 (SPSS Inc./IBM, Chicago, IL, USA).

## 3. Results

### 3.1. Prevalence of NPM1 and FLT3 Mutations

A total of 366 cases of AML were analysed. The patients’ characteristics are summarized in Appendix A. The median age was 51 years (range 15–74) and 203 (55.4%) were males. *NPM1* or *FLT3* mutations or both were detected in 46.7% of patients. The frequency of the *NPM1* mutation was 40.3%. The *FLT3* mutations were present in 27.8% (ITD: 18.4%; TKD: 10.4%). Both mutations (*NPM1* and *FLT3*) were present in 19.3% of patients. 

### 3.2. Study Cohort: Characteristics of Patients

According to the NILG AML 00/01 protocol, 171 patients with the *NPM1* mutation, *FLT3* mutations, or both were treated, and these represented the study cohort (Figure 1). The characteristics of this cohort are shown in Table 1.

Based on the molecular analysis for the *NPM1* and *FLT3*-ITD mutations, 27 (15.8%) patients had isolated *FLT3*-ITD (ITDm/*NPM1*wt), 38 (22.2%) concomitant *FLT3*-ITD, and *NPM1* mutations (ITDm/*NPM1*m), and 73 (42.7%) had an isolated *NPM1* mutation (*FLT3*wt/*NPM1*m). The *FLT3*-TKD mutations were present in 33 (19.3%) patients, 21 of whom were also *NPM1*-mutated (TKDm/*NPM1*m). 

The median age of the population studied was 54.5 years, without differences between the molecular subgroups. There were 32 patients aged over 64 years (19%). Most of them (28/32; 87.5%) showed an *NPM1* mutation (14 isolated and 14 associated with *FLT3* mutations).

In eleven patients, an extramedullary disease was documented, nine of whom were *NPM1*-mutated. The sites were the skin (ten), central nervous system (two), and nasopharynx (one), (two patients had both CNS and skin), without differences between the molecular groups. 

According to the protocol criteria, 26 patients received allo-SCT in their first CR (CR1), after a median of 5 months, of which 18 (69.2%) were *FLT3*-ITD-mutated (m) (Table 2). 

### 3.3. Response and Toxicities

After the first induction cycle, CR was achieved in 151/171 patients (88.3%) and 11 further patients obtained CR after two courses (162/171, 94.7%). The sixty-day treatment-related mortality was 2/171 (1.1%), due to uncontrolled infections. 

Table 2 shows the outcome of the *NPM1* or *FLT3* patients according to the molecular subgroups.

Overall, the CR rate after the first induction was significantly higher in *NPM1*m patients compared to *NPM1*wt [124/132 (93.9%) vs. 27/39 (69.2%): Fisher’s exact test, *p*: 0.0001)], whereas it was significantly lower in ITDm/*NPM1*wt compared to other subgroups [18/27 (66.7%) vs. 133/144 (92.4%): Fisher’s exact text *p*: 0.0009]. In TKDm, the CR rate was 90.9%, significantly higher in patients also with the *NPM1* mutation (21/21 in *NPM1*m vs. 9/12 in *NPM1*wt: Fisher’s exact test *p*: 0.04).

The *FLT3*-ITD allelic ratio was available in 28 patients; it did not influence the CR rate (95% and 87.5% for patients with a low or high allelic, respectively; Fisher’s exact test *p*: 0.49).

An *NPM1* MRD assessment was available in 62/151 of the CR patients (41%). After the first consolidation, 35/62 (56.4%) of the patients had achieved *NPM**1*-MRD negativity in their peripheral blood, with no difference between *NPM1*m patients with or without a concomitant ITD mutation (26/46, 56.5%, and 9/16, 56.2%, respectively). 

Overall, most patients were able to fully receive the pre-planned dosage of the consolidation course, and only seven patients have not completed the program. After the consolidation course with HD-ARAC, neutrophil recovery occurred a median of 10 days (6–17) after CD34+ cells reinfusion. The extra-hematological toxicities were gastrointestinal (nausea, stomatitis, and diarrhea) of grades 2 or 3 WHO, in 42% of the courses and neurological in two patients (one reversible cerebellar dysfunction with a grade 3 WHO and one reversible motor neuropathy with a grade 2 WHO). No cardiac side effects were observed. This intensive chemotherapy program was, overall, well-tolerated; the reinfusion of small amounts of autologous PBSC after each course of high-dose therapy may have potentially contributed to reducing toxicity. 

In summary, the CR rate after the ICE course was high in the whole population (88.3%), significantly higher in *NPM1*m patients, and lower in the ITD-isolated, without differences according to the allelic ratio. The treatment program was well-tolerated.

### 3.4. Relapse: Relapse Rate (RR) and Relapse-Free Survival (RFS)

After a median follow-up of 63.3 months (CI 95%: 51.6–100 months), 77 patients relapsed (52.1%).

The relapse rate was lower in patients with the *NPM1* mutation [54/127 (44.6%) vs. 23/35 (65.7%); *p*: 0.021], while it was higher among isolated ITDm patients if compared to patients with either co-mutated or isolated *NPM1*m (70.8%, 47.4%, 36.7%, respectively, *p*: 0.015). 

The timing of relapse differed between *FLT3*-ITD patients and *NPM1*-mutated patients, as 41% (7/17) of the ITDm/*NPM1*wt patients relapsed within 6 months and 88% within 12 (15/17) months compared to only 11% (3/26) of the *NPM1*m/*FLT3*wt within 6 months and 35% (9/26) within 12 months. 

*NPM1* MRD positivity in the peripheral blood after consolidation impacted their relapse adversely [MRD-negative 10/35 (28.6%) and MRD-positive 16/27 (59.2%); *p*: 0.02], particularly in the subgroup with an *NPM*1m/ITDm co-mutation (relapse risk: 85.7% in the MRD-positive vs. 11% in the MRD-negative, *p*: 0.0087).

Among patients who achieved CR, the median RFS was 42 months. At 3-years (y), the RFS was 52% +/− 8 (SE) (Figure 2a), without significant differences according to age [3-y RFS 45% +/− 10 (SE) in the older vs. 61% +/− 4.5 (SE) in the younger; *p*: 0.13].

RFS was not significantly different among the *NPM1*-mutated, with or without concomitant *FLT3*-ITD [3-y survival 67% +/− 5 (SE) and 50% +/− 8 (SE), respectively, *p*: 0.09] (Figure 2b). The allelic burden did not impact their RFS (*p*: 0.5), whereas patients with isolated *FLT3*-ITD had a significantly lower RFS (3-y 24%, *p*: 0.0002) (Figure 2b).

In TKDm patients, RFS was intermediate between the *NPM1*m and ITDm patients, and it was not influenced by the presence of the *NPM1* mutation [3-y RFS: 41% +/− 14 (SE) and 47% +/− 11 (SE) in the *NPM1*wt and in *NPM1*m patients, respectively; *p*: 0.9].

In summary, the relapse rate was lower in *NPM1*m patients, as opposed to ITDm-isolated patients, where it was higher and earlier. In *NPM1*m patients, the peripheral blood MRD positivity adversely impacted their relapse. The RFS was better in the *NPM1m* than in the *FLT3*-ITDm patients (*p* < 0.0001); a trend towards statistical significance was also observed in patients showing both mutations (*FLT3*-ITDm/*NPM1*m) (*p*: 0.09).

### 3.5. Overall Survival

In the whole cohort, sixty-nine out of one hundred and seventy-one (40%) patients died, fifty-four of disease progression, two of other causes (lung neoplasia and Amyotrophic Lateral Sclerosis), and ten from treatment-related causes (three of infection and seven from transplant-related mortality). Three patients developed a fatal secondary myelodysplastic disease, at 6, 11, and 24 months after their AML diagnosis, respectively.

The median survival was undefined and the OS at 3 and 5 years was 66% +/− 3 (SE) and 58% +/− 4 (SE), respectively. At 10 years survival, it was 51.5% +/− 4.6 (SE) (Figure 3a).

According to the molecular groups, in *NPM1*m patients, the median survival was undefined [OS at 3y and 5y: 72.4% +/− 4 (SE) and 63.6% +/− 5 (SE), respectively], significantly better than in the *NPM1*wt patients [median survival: 32 months, OS at 3y and 5y: 49.7% +/− 7(SE) and 43.6% +/− 7 (SE), respectively; *p*: 0.0017]; while in the *FLT3*m patients, the median survival was 57 months (OS at 3 and 5y: 55% +/− 5 and 49% +/− 5, respectively) significantly worse compared to the *FLT3*wt patients (*p*: 0.001).

Concerning the *FLT3*-ITD and *NPM1* reciprocal mutation status (Figure 3b), the median survival of the patients with *FLT3*-ITDm/*NPM1*wt was 18.5 months [OS at 3y and 5y: 32% +/− 8.8 (SE) and 23% +/− 8 (SE), respectively], significantly worse compared to the other subgroups (*p*: <0.0001 compared to the *FLT3-*ITDwt/*NPM1*m and *p*: 0.0076 compared to the *FLT3*-ITDm/*NPM1*m). Particularly, the median survival in patients with both mutations (*FLT3*-ITDm/*NPM1*m) was 98 months [OS at 3y and 5y 60% +/− 8 (SE) and 56% +/− 8 (SE), respectively], worse than in the *FLT3*-ITDwt/*NPM1*m, with a borderline significance (*p*: 0.06) [in the ITDwt/*NPM1*m: the median survival was undefined, at 3y and 5y OS 80% +/− 4.8 (SE) and 71% +/− 6 (SE), respectively]. The allelic ratio of the *FLT3*-ITD had no impact on the survival rates (*p*: 0.16).

In the TKDm patients, the median survival was 93.2 months, intermediate between the *NPM1*m-isolated and ITDm and without differences between the other group (*p*: 0.075 if compared to the ITD and *p*: 0.3 if compared to the *NPM1*m).

In the univariable and multivariable analysis, patients aged < 60 years and with the presence of *NPM1* mutations were correlated with significantly better survival (*p* 0.019 and *p* < 0.0001, respectively), while the presence of *FLT3*-ITD correlated with significantly worse survival (*p* < 0.0001).

Analysing the *NPM1*m and *FLT3*-ITD patients according to age, an *FLT3*-ITD mutation indicated poor survival in the younger patients (<65 years; *p*: 0.004), but it had no effect in the older patients (>66 years; *p*: 0.08), whereas *NPM1*m indicated better survival in both the older patients and in the younger patients (with any age cut off considered, <55 years, <60 years, and <65 years; *p* 0.000). In *FLT3*-ITDm/*NPM1*m patients, the survival was not impacted by age.

Allo-SCT was performed in 61 patients, in 26 in their CR1, according to protocol, and in 35 after haematological or molecular relapse. The median survival was 78 months, without differences between CR1 and CR2 (second CR) patients (5y-OS 55.7% +/− 10 (SE) in CR1 vs. 56.7% +/− 10 (SE) in CR2; *p*: 0.2). It was similar to the OS of non-transplanted patients [59.7% +/− 4 (SE) at 5y]. Overall, in our series, 44% (12/27) of all *FLT3*-ITD-mutated AML and only 10% (6/59) of co-mutated (*NPM1*m/*FLT3*m), underwent allo-SCT in their CR1.

In summary, the *NPM1* mutation correlated with significantly better survival, while the ITD mutation with significantly worse survival, as confirmed in the multivariable analysis. In co-mutated patients (*FLT3*-ITDm/*NPM1*m), the outcome was intermediate: worse than in *FLT3*-ITDwt/*NPM1*m, with a borderline significance (*p*: 0.06), but significantly better than *FLT3*-ITDm/*NPM1*wt (*p*: 0.0076).

## 4. Discussion

In this monocentric, real-life study, AML patients carrying the two most frequent mutations, *FLT3* and or *NPM1*, homogeneously treated over a period of 17 years with an aggressive idarubicin-based induction and the integration of idarubicin in the HD-ARAC consolidation, achieved an outstanding CR rate of 88.3% after one course of induction (ICE) and overall survival at 5 years of 58%. These results are similar to those obtained with the same NILG program in a multicentric setting, where in standard risk patients the CR rate after ICE was 82% and the 5y-OS 60% [26].

The choice of idarubicin in the induction instead of daunorubicin, and more importantly, its addition also to the HD-ARAC consolidation courses, may have played a role in improving the results of the treatment. It has been shown that the use of mitoxantrone and idarubicin instead of daunorubicin, during both the induction and consolidation in adult patients with AML who do not receive an allogeneic SCT, enhanced the long-term efficacy of chemotherapy, reducing their relapse risk [27]. A metanalysis of randomized trials comparing idarubicin with daunorubicin reported a survival advantage in AML patients treated with idarubicin [28]. Also, it has been shown that the combination of anthracyclines with high doses of cytarabine rather than their use as single agents or at lower doses may enhance the in vitro antileukemic T-cell immune response [29].

The optimization of antileukemic chemotherapy in our treatment program may explain the excellent outcome for *NPM1*-mutated patients. Indeed, the cytoplasmic delocalization of mutated *NPM1* reduces the anti-apoptotic activity of the *NPM1* protein, increases genomic instability, and may favour an increased sensitivity to high levels of cytotoxic agents [30]. In *NPM1*-mutated patients, their CR rate and survival at 3y and 5y were 93.9%, 72%, and 64%, respectively. These results compare favourably with the historical data of conventional chemotherapy regimens (CR 70% and 3y-OS 72% [13]; CR 91% and 5y-OS 49% [14]) and even with those obtained with the addition of Gemtuzumab Ozogamicin to the 3+7 regimen (CR 88.9%, 2y-OS: 68.3%, [31]) or with the use of a high-dose of cytarabine in the induction in the FLAG-Ida regimen (CR 90% and 3y-OS 63.5% [30]).

Monitoring of the *NPM1*-mutant measurable residual disease (MRD) allowed us to identify patients considered at an increased risk of relapse, as reported in [32,33]. In our study, *NPM1* MRD positivity in the peripheral blood after consolidation impacted relapse, particularly in the subgroup of *NPM1*m/ITDm AML. In our series, the strategy of using MRD positivity in CR is an indication that allo-SCT may have improved the results, allowing us to better stratify patients to be addressed for transplant. Therefore, the MRD evaluation should be included in all prospective studies and guide the clinician’s decision.

The very good long-term outcome of *NPM1*-mutated patients was adversely impacted neither by the presence of the *FLT3*-ITD or TKD mutation nor by their allelic burden. In these subgroups, the RATIFY trial proved that the addition of midostaurin, a first- generation *FLT3*-inhibitor, to the standard 3+7 induction regimen, improved the OS compared to placebo in patients aged <60 years of age, regardless of their allelic burden (AR ≤ 0.7 or > 0.7) or the type of mutation (ITD or TKD) (the median OS of 74.7 months in patients receiving midostaurin vs. 25.6 months in patients receiving placebo plus chemotherapy [18,20]). The benefit was confirmed in the different ELN risk categories. The 5-y OS of patients treated with midostaurin were 73%, 52%, and 43% in favourable, intermediate, and adverse ELN risk, respectively [19]. In the present study, without the addition of midostaurin, the median survival and the 5y-survival of the *NPM1* and *FLT3* double-mutated patients were 98 months and 56% +/− 8 (SE) in the ITD-mutated and 93 months and 70.8% +/− 11 (SE) in the TKD-mutated patients. These results do not compare unfavourably, although indirectly, with those obtained in the RATIFY trial (5y-OS: in TDKm, 70.5% and in ITDm, 63% low AR and 45% high AR). Interestingly, in an Australian real-life study, which included idarubicin in the therapeutic program, the results were better when compared with those obtained in the placebo arm of the RATIFY study. These data suggest that trying to optimize the treatment backbone to which newly available targeted agents are added could be an important step to further improve treatment results in AML [34]. Recently, an international expert panel updated the ELN guidelines, classifying *FLT3*-mutated AML in the intermediate group, independently of *NPM1* mutations’ status and allelic ratio. Our series seem to confirm an intermediate prognosis in patients with both the *NPM1* and *FLT3*-ITD mutation, which does not imply allo-SCT as a mandatory consolidation [35].

Despite the small number of patients, the results in *NPM1*-wt *FLT3*-ITD-mutated patients were unsatisfactory, with a lower CR and higher early relapse rates. In this subset of patients, where allo-SCT in their CR1 is universally recognized as the consolidation treatment of choice, the increased intensity of our chemotherapy program before allo-SCT did not significantly improve the outcomes, as some patients (3/27: 11%) never achieved complete remission, even after two induction courses, while others (9/27: 33.3%) relapsed early and failed to undergo transplantation. Indeed, for patients harbouring the *FLT3*-ITD mutation, as the pivotal AML mutation, the recently approved, highly selective, second-generation *FLT3* inhibitors may represent the optimal bridge to transplant, as recently demonstrated in the QuANTUM-First trial with the addition of quizartinib to a 3+7 induction treatment [36]. The new drugs would obtain a deeper remission, allowing for the gain of more time to proceed to consolidation with allo-SCT before relapse.

Although data on the prognostic significance of the *FLT3* TKD mutation are still controversial, the limited efficacy of our treatment program on *FLT3*-mutated patients emerged also in this subgroup of patients. The CR, relapse rate, RFS, and OS were intermediate between the ITD- and *NPM1*-mutated. Thus, the addition of targeted agents should also be pursued in the subgroup of patients with an *FLT3*-TKD-isolated mutation, as recently suggested by the data shown by Voso et al. [19].

The limitations of the present study are its retrospective nature and the limited availability of data on the *FLT3* allelic ratio and on *NPM1*-MRD, which derive from its very long observation time, which spanned periods when these techniques were not yet available.

## 5. Conclusions

The good outcome achieved without frontline allo-SCT in the group of *NPM1*-mutated patients, irrespective of their *FLT3* mutational status, suggests that the use of idarubicin, both in the induction and in consolidation, and of HD-ARAC plus limited autologous stem cell support, may reduce the need for early allo-SCT consolidation in the whole group of non-high-risk patients, including *NPM1*-mutated/high-burden *FLT3*-ITD patients. In addition, in *NPM1* patients, MRD monitoring PCR based during treatment may help to identify patients with a suboptimal response or in the early molecular relapse, allowing pre-emptive strategies of salvage therapy and more focused use of allo-SCT consolidation. Postponing allo-SCT after haematological or molecular relapse did not affect the transplant outcome in our series since the results were similar to patients undergoing allo-SCT in their first CR.

The results of this program could provide a benchmark for the evaluation of the additional benefit of incorporating second-generation *FLT3* inhibitors in effective chemotherapy programs in patients with *FLT3* and *NPM1* mutations.

## Figures and Tables

**Figure 1 cancers-14-04716-f001:**
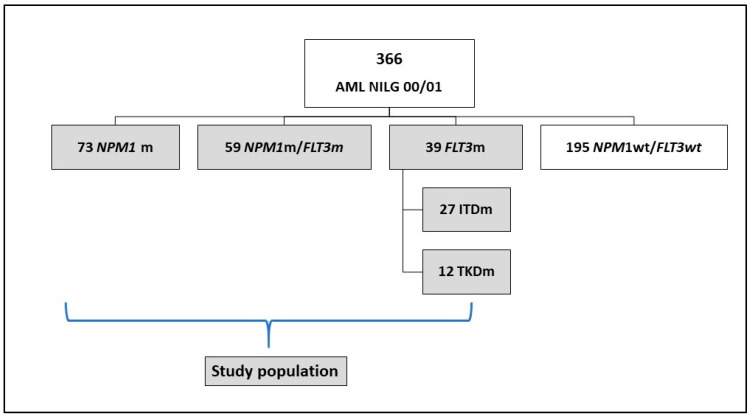
Study Cohort.

**Figure 2 cancers-14-04716-f002:**
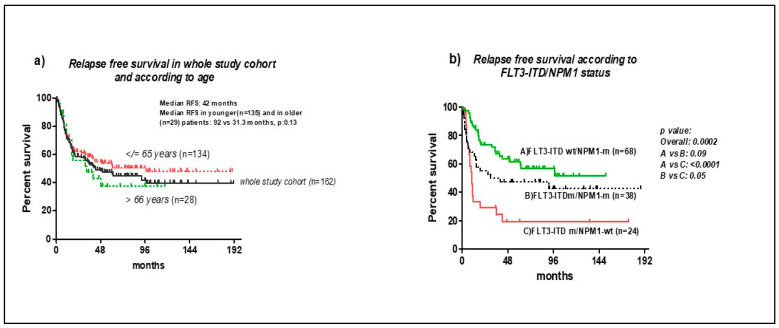
Relapse-Free Survival (RFS): (**a**) in the whole study cohort and according to age. (**b**) According to molecular data (*NPM1*/*FLT3*-ITD): *NPM1*m/ITDwt vs. *NPM1*wt/ITDm, vs. *NPM1*m/ITDm.

**Figure 3 cancers-14-04716-f003:**
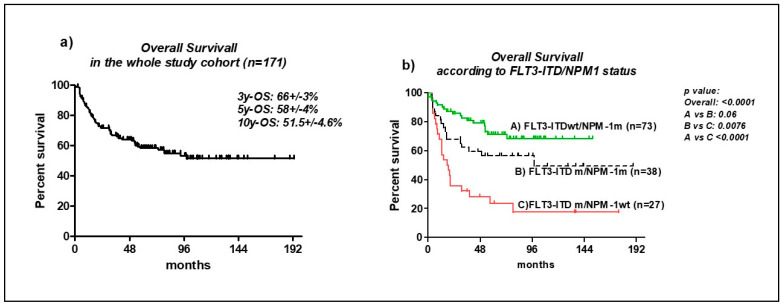
Overall Survival (OS): (**a**) in the whole study cohort; (**b**) According to molecular data: *NPM1/FLT3*-ITD status.

**Table 1 cancers-14-04716-t001:** Characteristics of 171 *NPM1*or *FLT3*-mutated patients (study cohort), divided according to molecular data.

	ITDm/*NPM*1wt n 27	ITDm/*NPM*1m n 38	TKDm/*NPM*1wt n 12	TKDm/*NPM*1m n 21	*FLT3*wt/*NPM*1m n 73	Total n 171
Median age (years)	55	55	54	54.5	55	54.5
Male n (%)	15 (55)	16 (42.1)	7 (58)	13 (61.9)	35 (47.9)	86 (50.3)
Extramedullary disease n (%)	0	1 (2.6)	2 (16)	2 (9.5)	6 (8.2)	11 (6.4)
Favorable K n (%)	1 (3.7)	-	2 (16.7)	-	-	3 (1.8)
Intermediate K n (%)	23 (85.2)	38 (100)	8 (66.6)	21 (100)	71 (97.3)	161 (94.2)
Adverse K n (%)	3 (11.1)	-	2 (16.7)	-	2 (2.7)	7 (4)

Abbreviations: n: number, K: karyotype; m: mutated, and wt: wild-type.

**Table 2 cancers-14-04716-t002:** Complete Response, duration of CR, relapse rate, RFS, and OS, according to molecular groups.

	ITDm/*NPM1*wt n (%) 27 (15.8)	ITDm/*NPM1*m n (%) 38 (22.2)	TKDm/*NPM1*wt n (%) 12 (7)	TKDm/*NPM1*m n (%) 21 (12.3)	*FLT3*wt/*NPM1*m n (%) 73 (42.7)	Total n 171
CR rate after ICE	18 (66.7)	37 (97.3)	9 (75)	21 (100)	66 (90)	151 (88.3)
CR rate after 2 courses	24 (88.9)	38 (100)	11 (92)	-	68 (93.2)	162 (94.7)
Relapse rate	17 (70.8)	18 (47)	6 (54.5)	11 (52.4)	25 (36.7)	77 (52.1)
Duration CR <6 months	6 (25)	8 (21)	1 (9)	2 (9.5)	2 (2.9)	19/162 (11.7)
Median duration CR (months)	11	35	15.9	19.7	48	32.5
Median RFS (months)	10	35	32	33	undef	42
Median OS (months)	18.46	98	undef	93	undef	undef
5y-OS (% +/− SE)	23 +/− 8	56 +/− 8	54 +/− 15	70.8 +/− 11	71 +/− 6	58 +/− 4
Allo-SCT in CR1 during disease course	15 12 (80) 3	11 6 (67) 5	7 4 (57) 3	6 - 6	22 4(18) 18	61 26 (42.6) 35 (57)
Median RFS (months) censored at allo-SCT	10	undef	11.5	48	undef	undef
Median OS (months) censored at allo-SCT	10	undef	31	93	undef	undef
5y-OS (% +/− SE) censored at allo-SCT	16.7 +/− 11	57 +/− 9	40 +/− 21	68 +/− 13	75 +/− 7	60 +/− 4

Abbreviations: m: mutated, wt: wild-type, Allo-SCT: allogenic stem cell transplantation, CR: complete remission, CR1: first complete remission, SE: standard Error, OS: overall survival, y: years, and RFS: relapse-free survival.

## Data Availability

Data available on request due to restrictions, e.g., privacy or ethical.

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
