# Peer review of "Prognostic Relevance of NPM1 and FLT3 Mutations in Acute Myeloid Leukaemia, Longterm Follow-Up—A Single Center Experience"

_cancers, 2022, doi:10.3390/cancers14194716_

Round 1

Reviewer 1 Report

IIn the paper Borlenghi and coworkers describe the outcome of NMP1 and FLT3 mutated AML patients treated in a single center. The compared different treatment outcome based on the FLT3/NPM1 mutation profile.

The paper has some interesting findings, although I have some concerns.

Regarding patients proceeding to allo-SCT in CR, it seems quite low, with only 26 patients. It also unclear in which genetic groups these patients were belonging, and this could be stated in Table 2. The data is maybe too low to tell about the impact of allo-SCT, although it was little unclear for me how the authors recommend an allo-SCT, and if they think more patients should have been offered an allo-SCT in CR1.

A limitation of the study is the lack of FLT3 inhibitors used, which is currently remanded, and how this should be implemented in further treatment. Furthermore, the ELN guidelines has recently been updated (Diagnosis and Management of AML in Adults: 2022 ELN Recommendations from an International Expert Panel, Dønher at al, Blood Epub ahead of print). Here FLT3 mutated AML is classified in the intermediate group, independently of NPM1 mutations status and allelic ration. How does this fall in accordance with the authors findings in the present study? Should FLT-3-ITD/NPM1mut patient be transplanted in CR1 according to the authors?

The use of auto-SCT in the study was also unclear for med, do the author think this has a future palace in AML treatment?

The use of MRD is also of some limitations in the study. How to the authors think this should be used in further AML studies?

Genetic names should be written in italic.

Author Response

Reviewer 1.

 1)In the paper Borlenghi and coworkers describe the outcome of NMP1 and FLT3 mutated AML patients treated in a single center. The compared different treatment outcome based on the FLT3/NPM1 mutation profile.The paper has some interesting findings, although I have some concerns.Regarding patients proceeding to allo-SCT in CR, it seems quite low, with only 26 patients. It also unclear in which genetic groups these patients were belonging, and this could be stated in Table 2. The data is maybe too low to tell about the impact of allo-SCT, although it was little unclear for me how the authors recommend an allo-SCT, and if they think more patients should have been offered an allo-SCT in CR1.

We agree with the Reviewier concerning the low number of patients proceeding to allotransplant. According to the protocol criteria, allo-SCT was planned in patients with FLT3-ITD isolated mutation, adverse karyotype or late responder (CR after 2 courses), in patients with persistent molecular disease detectable at the end of the entire treatment program and in patients with haematological or molecular relapse (as reported in Methods, line 125 of manuscript). In our study, of 26 patients proceeding to allo-SCT in 1 CR, 12 (46%) were FLT3-ITD mutated without NPM1 mutation. 15/27 patients FLT3-ITD mutated, candidates to allo-SCT in first CR, did not proceed to allo-sct because of refusal (3) and refractory disease or early relapse while waiting for a matched unrelated donor (12).

We have better explainedthis concept by expanding the “Discussion” section (line 312).

2) A limitation of the study is the lack of FLT3 inhibitors used, which is currently remanded, and how this should be implemented in further treatment. Furthermore, the ELN guidelines has recently been updated (Diagnosis and Management of AML in Adults: 2022 ELN Recommendations from an International Expert Panel, Dønher at al, Blood Epub ahead of print). Here FLT3 mutated AML is classified in the intermediate group, independently of NPM1 mutations status and allelic ration. How does this fall in accordance with the authors findings in the present study? Should FLT-3-ITD/NPM1mut patient be transplanted in CR1 according to the authors?

Our series seem to confirm, the recent prognostic reclassification ELN 22, proposed by Doehner et al., which classifies patients with FLT3 mutated AML independently of NPM1 mutations status and FLT3 ITD allelic ratio, as intermediate.

We agree with the Reviewier that the lack of FLT3 inhibitors is a limitation of the study, which was  designed before their availability. However, the strategy used proved very effective in the subgroup of patients NPM1m/FLT3m with a median survival of 98 months and 5y OS of 56%. Even if the relapse rate (10 molecular and 8 haematological) was high (18/38: 47%),  MRD monitoring allowed to early identify patients candidates to allo-SCT

In our study only 6/38 (16%) patients proceeded to allo-SCT in CR1 (2 with extrahaematological disease and 4 for persistent MRD positivity) but four more patients received allo-SCT  for molecular relapse and 1 in CR2.

Therefore, we believe that in this subgroup allo-SCT is not mandatory as consolidation in first CR since  careful monitoring of MRD helps identifying the subgroup patients who deserve an intensification.

We have tried to better explain this concept in the “Discussion” section by adding a pertinent sentence (line 308)

3) The use of auto-SCT in the study was also unclear for med, do the author think this has a future palace in AML treatment?

In our study, the reinfusion of CD34+ peripheral blood stem cell (PBSC), was aimed to minimize haematological toxicity, due to the very high dose of ARAC delivered and associated with idarubicin, it was only an autologous stem cell support not an auto-SCT

In “Methods” section, (“2.3. Treatment plan, risk assessment and criteria of response”), we modified the sentence, line 122

4)The use of MRD is also of some limitations in the study. How to the authors think this should be used in further AML studies?

We agree with Reviewer, as in our study MRD evaluaton was available only in  41% of patients, but even in this small series of patients the evaluation of MRD has proven useful in patients’ management in the real-life setting and has been shown to have a prognostic role. Therefore, we believe that it should be included in all prospective studies and guide the clinician's decision. We added a sentence in the “Discussion” section (line 291)

5) Genetic names should be written in italic.

We thank the Reviewer for the observation and correctedthe text accordingly (in red the modifications).

Reviewer 2 Report

In this article, Erika et al reported clinical characteristics and outcomes of genetically distinct(NPM1, FLT3ITD) AML patients(n= 171) which are generally consistent with the literature. The authors urge that these results could serve as a benchmark for future trials that combine chemo and 2nd generation FLT3 inhibitors. While the information present is valuable, the reviewer found critical pieces of information missing and the overall flow is hard to follow, as detailed below.

*Clinical information is over-simplified, tables should include other coexisting mutations, blast counts, plt count, WBC, etc, and treatment. These pieces of information may significantly impact patient outcomes.

*Multivariate analysis shall be performed to thoroughly examine the impact of other factors eg. age, genders on the prognosis in each genetic group as they have been reported to indicate differential survival (Gunnar et al. Blood Adv 2020). 

 *Indicate the number of patients in each group in survival curve figures.

*It is unclear if these patients are de novo AML or not and what treatment they have received.

*When comparing and discussing characteristics of different genetics groups, the reviewer recommends the authors avoid making cross-subgroup comparisons (ITDm/NPM1wt vs ITDwt/NPM1m) before comparing the two major groups, NMP1 mutated and FLT3. All comparisons in each section should follow the same order otherwise it’s super difficult to follow.

*It is unclear if those patients who received allo-transplant are included in the figures. When comparing OS and RFS, patients who received allo-transplant should be separated from those who did not.

*The authors want to provide a benchmark from chemo for future evaluation of FLT3 inhibitors. Of note, the cohort in this study is relatively young and fit (median 51 years old) to qualify for high intensive chemo and seem lightly treated or de novo (which is unclear).

Line 75, “they suggest” is awkward. Consider change to “evaluation…” is “needed”

Line 180. It is quite surprising only 7 patients did not complete the program given the intensity of this regimen. The reviewer wonders if they are all aged individuals e.g. >70.

Line 215. Please be more specific about “secondary myelodysplastic disease” that is lethal.

Line 276, RATIFY

Author Response

Reviewer 2.

1) In this article, Erika et al reported clinical characteristics and outcomes of genetically distinct(NPM1, FLT3ITD) AML patients(n= 171) which are generally consistent with the literature. The authors urge that these results could serve as a benchmark for future trials that combine chemo and 2nd generation FLT3 inhibitors. While the information present is valuable, the reviewer found critical pieces of information missing and the overall flow is hard to follow, as detailed below.

*Clinical information is over-simplified, tables should include other coexisting mutations, blast counts, plt count, WBC, etc, and treatment. These pieces of information may significantly impact patient outcomes.

We agree with the Reviewer about the relatively low number of clinical informations, but the period of observation is very long (17 years) and we did not perform a comprehensive genetic study in all the patients at the beginning of the study. Moreover, unfortunately, the retrospective nature of the study does not allow us to recover all the clinical data, that would certainly contribute to get a more complete study.

We have emphasized this aspect in the “Methods” section, (“2.2 Diagnostic workup and Molecular and Cytogenetic analysis”) (line 111)

*Multivariate analysis shall be performed to thoroughly examine the impact of other factors eg. age, genders on the prognosis in each genetic group as they have been reported to indicate differential survival (Gunnar et al. Blood Adv 2020). 

We are grateful for this suggestion. We performed multivariate analysis and we analyzed age, gender, FLT3-ITD, FLT3-TKD, NPM1, karyotype. Age, NPM1 mutation and FLT3-ITD mutation proved to impact on survival.

We added a sentence in Methods Section, 2.4. Statistical analysis paragraph (line 144) and a sentence in “Results” Session, in 3.4 Overall Survival paragraph (line 245)

*Indicate the number of patients in each group in survival curve figures.

We modified the survival curve figures adding the number of patients, as requested

*It is unclear if these patients are de novo AML or not and what treatment they have received.

The study population included only de novo AMLas specified  in the Methods Section, Study Design and endpoints paragraph. Line 95

 *When comparing and discussing characteristics of different genetics groups, the reviewer recommends the authors avoid making cross-subgroup comparisons (ITDm/NPM1wt vs ITDwt/NPM1m) before comparing the two major groups, NMP1 mutated and FLT3. All comparisons in each section should follow the same order otherwise it’s super difficult to follow.

We are grateful for the suggestion. We modified the sentences in the “Results” Section to better clarify the data flow (line 197 and 229)

*It is unclear if those patients who received allo-transplant are included in the figures. When comparing OS and RFS, patients who received allo-transplant should be separated from those who did not.

The figures of OS and RFS included also patients that received allo-SCT (61 patients). Indeed, our series includes both patients undergoing allo-SCT in CR1 (whenever indicated, 26 patients) and patients undergoing allo-SCT after relapse or because of MRD persistence (36 patients). We are grateful for this suggestion, we performed the survival analysis also by censoring patients at allo-SCT. Results are added in table 2 (RFS and OS), and if it necessary we can also add the survival curves.

*The authors want to provide a benchmark from chemo for future evaluation of FLT3 inhibitors. Of note, the cohort in this study is relatively young and fit (median 51 years old) to qualify for high intensive chemo and seem lightly treated or de novo (which is unclear).

Our study was planned for patients younger than 65 years. Indeed, also the RATIFY study enrolled patients less than 60 years. We know that many AML patients are older than 65 years, but this protocol may be applicable also in elderly fit patients with chemotherapy dose adjustement whenever indicated (Appendix1, line 386).

 Line 75, “they suggest” is awkward. Consider change to “evaluation…” is “needed”

We thank, we modified as suggested (line 90)

Line 180. It is quite surprising only 7 patients did not complete the program given the intensity of this regimen. The reviewer wonders if they are all aged individuals e.g. >70.

Seven patients did not complete the therapeutic program: 3 patients because of aplastic death for infection (1, encephalitis, 53 yo; 1 sepsis, 45 yo; and 1 invasive fungal infection, 63 yo), 1 because of neuro-toxicity (60 yo) and 3 for prolonged cytopenia evolving into myelodysplasia (58, 60, 62 yo). This intensive chemotherapy program was overall well tolerated, probably because of the availability of small amount of autologous stem cell, which were reinfused after high dose chemotherapy.

We thank for this observation, we added a sentence in “Results” session (Response and toxicities) (line 192)

Line 215. Please be more specific about “secondary myelodysplastic disease” that is lethal.

We modified the sentence to better explain (line 225)

Line 276, RATIFY

Thank you, we corrected the name of study (line 305)

Round 2

Reviewer 2 Report

The reviewer's comments/suggestions have been adequately addressed.

Given the amount of information presented and the readers are likely fellow physicians, the reviewer recommends adding a one-sentence summary/take-home message for each paragraph. And ask one of your hema/oncology residents to read this manuscript and see if he/she can get them.

Author Response

Comments from reviewer 2

Given the amount of information presented and the readers are likely fellow physicians, the reviewer recommends adding a one-sentence summary/take-home message for each paragraph. And ask one of your hema/oncology residents to read this manuscript and see if he/she can get them.

We are grateful for this suggestion, we added a sentence at the end of most paragraphes in the Results Section [3.3 Response and toxicities, 3.4 Relapse: Relapse Rate (RR) and Relapse Free Survival (RFS) and 3.5 Overall Survival]